# ANCOC Score to Predict Mortality in Different SARS-CoV-2 Variants and Vaccination Status

**DOI:** 10.3390/jcm12185838

**Published:** 2023-09-08

**Authors:** Marcello Candelli, Marta Sacco Fernandez, Giulia Pignataro, Giuseppe Merra, Gianluca Tullo, Alessandra Bronzino, Andrea Piccioni, Veronica Ojetti, Antonio Gasbarrini, Francesco Franceschi

**Affiliations:** 1Emergency, Anesthesiological and Reanimation Sciences Department, Fondazione Policlinico Universitario A. Gemelli—IRCCS of Rome, 00168 Rome, Italy; giulia.pignataro@policlinicogemelli.it (G.P.); alessandra.bronzino@policlinicogemelli.it (A.B.); andrea.piccioni@policlinicogemelli.it (A.P.); veronica.ojetti@policlinicogemelli.it (V.O.); francesco.franceschi@policlinicogemelli.it (F.F.); 2Department of Emergency Medicine, Università Cattolica del Sacro Cuore of Rome, 00168 Rome, Italy; marta.sacco01@icatt.it (M.S.F.); gianluca.tullo01@icatt.it (G.T.); 3Biomedicine and Prevention Department, Section of Clinical Nutrition and Nutrigenomics, Facoltà di Medicina e Chirurgia, Università degli Studi di Roma Tor Vergata, 00133 Rome, Italy; giuseppe.merra@uniroma2.it; 4Medical, Abdominal Surgery and Endocrine-Metabolic Science Department, Fondazione Policlinico Universitario A. Gemelli—IRCCS of Rome, 00168 Rome, Italy; antonio.gasbarrini@policlinicogemelli.it

**Keywords:** prognostic score, COVID-19, SARS-CoV-2, mortality

## Abstract

Background: More than three years after the severe acute respiratory syndrome coronavirus 2 (SARS-CoV-2) pandemic outbreak, hospitals worldwide are still affected by coronavirus disease 19 (COVID-19). The availability of a clinical score that can predict the risk of death from the disease at the time of diagnosis and that can be used even if population characteristics change and the virus mutates can be a useful tool for emergency physicians to make clinical decisions. During the first COVID-19 waves, we developed the ANCOC (age, blood urea nitrogen, C-reactive protein, oxygen saturation, comorbidities) score, a clinical score based on five main parameters (age, blood urea nitrogen, C-reactive protein, oxygen saturation, comorbidities) that accurately predicts the risk of death in patients infected with SARS-CoV-2. A score of less than −1 was associated with 0% mortality risk, whereas a score of 6 was associated with 100% risk of death, with an overall accuracy of 0.920. The aim of our study is to internally validate the ANCOC score and evaluate whether it can predict 60-day mortality risk independent of vaccination status and viral variant. Methods: We retrospectively enrolled 843 patients admitted to the emergency department (ED) of our hospital with a diagnosis of COVID-19. A total of 515 patients were admitted from July 2021 to September 2021, when the Delta variant was prevalent, and 328 in January 2022, when the Omicron 1 variant was predominant. All patients included in the study had a diagnosis of COVID-19 confirmed by polymerase chain reaction (PCR) on an oropharyngeal swab. Demographic data, comorbidities, vaccination data, and various laboratory, radiographic, and blood gas parameters were collected from all patients to determine differences between the two waves. ANCOC scores were then calculated for each patient, ranging from −6 to 6. Results: Patients infected with the Omicron variant were significantly older and had a greater number of comorbidities, of which hypertension and chronic obstructive pulmonary disease (COPD) were the most common. Immunization was less common in Delta patients than in Omicron patients (34% and 56%, respectively). To assess the accuracy of mortality prediction, we constructed a receiver operating characteristic (ROC) curve and found that the area under the ROC curve was greater than 0.8 for both variants. These results suggest that the ANCOC score is able to predict 60-day mortality regardless of viral variant and whether the patient is vaccinated or not. Conclusion: In a population with increasingly high vaccination rates, several parameters may be considered prognostic for the risk of fatal outcomes. This study suggests that the ANCOC score can be very useful for the clinician in an emergency setting to quickly understand the patient’s evolution and provide proper attention and the most appropriate treatments.

## 1. Introduction

More than three years after the outbreak of the severe acute respiratory syndrome coronavirus 2 (SARS-CoV-2) pandemic, the impact on hospitals worldwide is still significant [1]. Patients with coronavirus disease 19 (COVID-19) exhibit remarkable clinical variability, ranging from completely asymptomatic forms to severe clinical pictures with acute respiratory failure and death [2]. As the pandemic progressed, new virus variants continued to emerge. After the wild-type virus, a whole series of variants were isolated [3]. It was reported that some of them spread over time and became predominant in successive pandemic waves [3]. The Alpha, Beta, Delta, and Omicron variants (with numerous subvariants) had the greatest impact. In particular, the Delta variant had higher infectivity and pathogenicity and was associated with poorer prognosis and higher mortality than the Alpha and Beta variants [4]. The Omicron variant, while more infectious than the Delta variant, had lower pathogenicity, lower mortality, and better clinical outcome [5]. Since the SARS-CoV-2 pandemic outbreak, several prognostic factors have been recognized to stratify the risk of severe illness or death [6]. This risk stratification is essential because it can subsequently lead to optimized treatment and resource utilization in the care of these patients. Risk prediction scores are important tools to support clinical decision making in patients with COVID-19. [7]. Although many scores have been used and validated in clinical research for risk stratification (need for intensive care unit and death), we do not have data on their use in the real world and they are not included in international guidelines and recommendations from international health organisations.

Existing scores were studied at the onset of the pandemic; however, new scores have been developed and validated over time. In a study conducted at our hospital, the national early warning score (NEWS), the NEWS2, the modified NEWS (NEWS-C), the modified early warning score (MEWS), the quick sequential organ failure assessment (qSOFA), and the rapid emergency medicine score (REMS) were calculated retrospectively on arrival for all patients who had COVID-19 [8]. It was found that NEWS and REMS, measured at the emergency department (ED) visit, were the most sensitive predictors of 7-day admission to the intensive care unit or death [8]. On the other hand, new and specific scores have been developed to assess the prognosis of COVID-19 [9]. One of the most commonly used is the 4C mortality score, which evaluates eight variables (age, sex, concomitant diseases, respiratory rate, peripheral oxygen saturation, state of consciousness, urea level, and C-reactive protein) and has a range of 0 to 21 points [10]. Our research group determined another prognostic score, the GOL2DS score (blood group A, the ratio between the partial pressure of oxygen in arterial blood and the inspired oxygen fraction [PaO_2_/FiO_2_], lactate dehydrogenase [LDH], lactate, dyspnea, and oxygen saturation [SpO_2_]), which indicates the likelihood that patients infected with SARS-CoV-2 will need to be admitted to an intensive care unit (ICU) [11]. In a previous study, conducted in March 2020, we developed the ANCOC score based on five parameters: age, blood urea nitrogen, C-reactive protein, oxygen saturation, and comorbidities. It ranges from −6 to +6 and correlates proportionally with the patient’s 60-day mortality risk [12]. Compared to the previously mentioned scores, the latter, therefore, seems to be easier to apply in an emergency. However, many things have evolved since the beginning of the COVID-19 pandemic, including the virus itself (we have seen countless mutations of the virus) and the affected population (which is increasingly vaccinated) [13]. The aim of our study was to validate the diagnostic accuracy of the ANCOC score in the presence of vaccination and a high prevalence of different SARS-CoV-2 variants.

## 2. Materials and Methods

We retrospectively enrolled 843 patients who visited the ED of the Fondazione Policlinico Universitario Agostino Gemelli—IRCCS (Istituto di Ricerca e Cura a Carattere Scientifico) in Rome from July 2021 to September 2021 (when the Delta variant was predominant) [11] and in January 2022 (when the Omicron 1 variant was predominant) [14,15] with COVID-19. A total of 515 patients were enrolled during the Delta wave and 328 patients during the Omicron wave.

SARS-CoV-2 infection was confirmed by polymerase chain reaction (PCR) on an oropharyngeal swab. The study was performed in accordance with the Declaration of Helsinki and the study protocol was approved by the ethics committee of the Catholic University of the Sacred Heart of Rome (ID: 4916).

Demographic data, comorbidities, and various laboratory, and blood gas parameters were obtained from all patients. Of the patients initially included, we excluded patients with mixed infections (i.e., COVID-19 and other infectious diseases) and patients with COVID-19 and other acute conditions affecting prognosis (i.e., surgical emergencies, time-dependent diseases, pathologies with high risk of fatal outcome).

We also divided patients into two groups according to which variant they were more likely infected with (Delta or Omicron 1).

The following data were collected for each patient using the electronic hospital record:-Demographic data (age; sex; body mass index (BMI); number and type of comorbidities);-COVID-19 infection and hospitalization data (first oropharyngeal swab positivity confirmed by PCR; cutoff index (COI) value of the rapid antigen test performed in the emergency department (ED); initiation of anticoagulant therapy at home or in the hospital; day of admission and duration of hospitalization; final outcome; intensity of hospitalization);-Therapy administered during hospitalization: pharmacologic therapy (corticosteroids, monoclonal antibodies, remdesivir, anti-interleukin 6 receptor) and oxygen therapy (high flow nasal therapy (HFNC); non-invasive ventilation (NIV); orotracheal intubation (OTI));-Laboratory data (blood urea nitrogen [BUN], CRP, procalcitonin, LDH, lymphocytes, eosinophils, D-dimer, fibrinogen);-FiO_2_, PaO_2_, PaO_2_/FiO_2_, SpO_2_ of the first blood gas analysis performed upon arrival at ED;-Immunization data: vaccinated (at least one booster) or not vaccinated;-We calculated the ANCOC score for all included patients (Table 1).

### 2.1. Statistical Analysis

Data were expressed as means ± SD for continuous variables and as numbers and percentages for dichotomous variables. Comparison between groups was performed with Student’s *t* test for continuous and normally distributed variables and with chi-square or Fisher’s test for dichotomous variables. Multivariate analysis (logistic regression) was performed between groups by entering all variables that had a *p* value of at least 0.1 in the univariate analysis and correcting for sex and age. To evaluate the accuracy of the ANCOC score in predicting the risk of death in the different patient groups, a receiver operating characteristic (ROC) curve was constructed and the area under the ROC (AUROC) curve was calculated. A *p* value <of 0.05 was considered significant.

### 2.2. Sample Size

The sample size was calculated to provide an estimated accuracy value of at least 0.8 for the ANCOC score using an AUROC curve. Considering the mortality of 15.2% observed in our previous study [12], a width of the confidence interval of 0.150, a confidence interval of 0.95, an alpha error of 0.05, and a power of 80%, we calculated a minimum sample size of 348 patients [16].

## 3. Results

Of the 843 patients included in this study, 481 were male (57%) and 362 were female (43%). The mean age of all enrolled patients was 62 ± 19 years (range 18–100) The mean age of patients infected with the Delta variant was 60 ± 19 years (range 18–100), while that of patients infected with the Omicron variant was 64 ± 18 years (range 20–96) (*p* = 0.00082). The mean BMI of included patients was 27.3 ± 6 kg/m^2^ (range 15.3–80.8). The mean BMI of patients infected with the Delta variant was 27.4 ± 7 kg/m^2^ (range 15.5–80.8), and that of patients infected with the Omicron variant was 27.1 ± 6 kg/m^2^ (range 15.3–60.1) (*p* = 0.6) (Table 1). Moreover, 515 patients were recruited from July to September 2021, a period when the Delta variant was predominant and the remaining 328 were recruited in January 2022, a period when the Omicron variant was predominant [14,15]. Patients differed by the presence or absence of comorbidities and by the number of comorbidities. Of the patients, 265 (31%) had no comorbidities, 218 (26%) had one comorbidity, and 360 (43%) had two or more comorbidities (Table 2). However, the differences in the number of comorbidities between groups were not statistically significant. The main comorbidities were also compared: diabetes, hypertension, ischemic heart disease, heart failure, atrial fibrillation, COPD, obesity, history of active neoplasia, Parkinson’s disease, Alzheimer’s disease, and other chronic diseases (Table 3). The prevalence of hypertension, obesity, chronic kidney disease, and COPD differ significantly between groups. However, in the multivariate analysis we found that there was a significant difference in prevalence between the waves studied only for hypertension and COPD. These two diseases occurred more frequently in patients infected with the Omicron variant than in patients infected with the Delta variant (Table 4).

The prevalence and mean values of the respiratory parameters studied were compared between patients infected with the Delta variant (July–September 2021 wave) and those infected with the Omicron variant (January 2022 wave). 

Patients infected with the Delta variant had a mean oxygen saturation of 93 ± 6 while those infected with the Omicron variant had a mean oxygen saturation of 94 ± 5 (*p* = 0.0001). Patients infected with the Delta variant had a mean PaO_2_/FiO_2_ of 296 ± 109 while patients infected with the Omicron variant had 324 ± 115 (*p* = 0.00093) (Table 5).

Evaluation of the patients’ laboratory parameters revealed differences between the two groups. The parameters associated with a difference in SARS-CoV-2 variant in univariate analysis, with *p* < 0.1 were blood urea nitrogen, LDH, eosinophils, and fibrinogen (Table 6).

We also investigated whether there were differences in treatment between the two groups: 139 patients (27%) with the Delta variant were treated with remdesivir and 55 patients (16%) with the Omicron variant (*p* = 0.005); 361 patients (70%) with the Delta variant were treated with corticosteroids and 147 patients (45%) with the Omicron variant (*p* = 0.0001). Only 17 patients in the Delta group and 9 patients in the Omicron group were treated with molnupiravir (*p* = 0.83). Finally, 113 patients (22%) with the Delta variant and 13 (4%) with the Omicron variant were treated with anti-IL6R (*p* = 0.001). No patients were treated with nirmaterlevir-ritonavir because it was not approved in Italy until the end of January 2022. No differences were observed between groups regarding anticoagulants. In the Delta group, 434/515 (84%) were taking anticoagulants, versus 263/328 (80%; *p* = 0.64) in the Omicron group,. Of these, 71/515 (14%) in the Delta group and 43/328 (13%) in the Omicron group were already taking anticoagulant therapy prior to diagnosis with COVID-19, while 363/515 (70%) and 220/328 (67%) started anticoagulant treatment during hospitalization. Among the hospitalized patients, enoxaparin was used as a prophylactic, non-weight-adjusted dosage (4000 IU once daily) in 73% of patients.

Most patients required oxygen supplementation. Of patients with the Delta variant, 88 (17%) required high-flow nasal cannula (HFNC) oxygen therapy, while only 29 patients (9%) with the Omicron variant required it (*p* = 0.001); 60 patients (11%) with the Delta variant and 29 patients (9%) with the Omicron variant were ventilated with non-invasive ventilation (NIV) (*p* = 0.01) (Table 7).

In addition, the two populations infected with the two different SARS-CoV-2 variants had differences in outcome. For example, 79 patients (15%) with the Delta variant were discharged compared to 122 patients (34%) with the Omicron variant (*p* = 0.0001). In the Delta population, 336 (65%) patients were hospitalized in a medical ward, compared with 173 (53%) in the Omicron population. Of the patients hospitalized in the ICU, 100 (20%) were in the Delta population and 43 (13%) were in the Omicron population (Table 8).

Regarding vaccination status, our data showed that among the Delta patients, 176 (34%) were vaccinated with at least one shot and 339 (66%) were not vaccinated. Of the vaccinated patients, 29 had one and 147 had two shots. In contrast, 183 (56%) versus 145 (44%) patients were vaccinated (*p* = 0.001) during the Omicron wave. The majority of patients (83%) received the mRNA-BNT162b2 vaccine, 11% received the mRNA-1273 vaccine, and 3% received both. For the remaining patients (3%), we did not have information on the type of vaccine used. Of vaccinated patients, 12 received one dose, 80 two doses, and 90 three doses (Table 9). 

The ANCOC score was then calculated for each patient, the ROC curves were generated, and the area under the curve (AUC) was assessed for the entire population and for subgroups (Delta variant, Omicron variant, vaccinated, and not vaccinated) to predict mortality outcomes. We found high accuracy of the ANCOC score in predicting death, with an AUROC of 0.841 (*p* < 0.0001) (Figure 1). We confirmed that an ANCOC score of less than 1 had a sensitivity of 95% (56% specificity) to exclude death at 60 days and an ANCOC score of more than 3 had a specificity of 99.9% for 60-day mortality. The high accuracy of the ANCOC score in predicting mortality in COVID-19 patients was found in all subgroups studied (Figure 2, Figure 3, Figure 4 and Figure 5).

## 4. Discussion

The aim of this study was to assess whether the ANCOC score studied during the first wave of COVID-19 is still useful in the era of vaccination and high prevalence of the new SARS-CoV2 variants. To determine whether our aim was justified, we first analyzed the differences between the Delta and the Omicron waves. First of all, patients differed in age: those infected with the Omicron variant were significantly older than those infected with the Delta variant, with a difference of approximately 5 years, which can be explained by the fact that the protection of the booster vaccine administered from November 2021 was effective [17]. Regarding comorbidities, there was a significant difference in prevalence between the studied waves only for hypertension and COPD, which were more frequent in patients with the Omicron variant. It is known from other studies that COPD patients have significantly increased expression of the ACE2 receptor, which is used by SARS-CoV-2 to enter epithelial cells [18]. This could explain why this group of patients is more prone to infections. On the other hand, patients with hypertension should theoretically have increased expression of ACE and decreased expression of ACE2, the latter being a protective enzyme against elevated blood pressure levels. However, this would not explain why hypertension is an important concomitant disease in SARS-CoV-2-infected patients, since ACE2 is the entry point [19]. A possible explanation could be the presence of single nucleotide polymorphisms (SNPs) in the ACE2 gene, which could lead to an increase in blood pressure in patients (because they are less able to counteract the effects of ACE) on the one hand and have an increased affinity for the SARS-CoV-2 spike protein and in particular that of the Omicron variant, on the other hand [20]. In our univariate analysis, we found lower eosinophil levels in patients with the Delta variant than in patients with the Omicron variant. There are previous studies in the literature that associate peripheral eosinopenia with a worse prognosis (assessed as mortality and need for intensive care) [21,22]. This finding is consistent with ours. It is now well established that the lethality of the Omicron variant is lower than that of the Delta variant [5,23], and a different inflammatory response involving eosinophils may explain this difference, at least in part. Indeed, eosinophils are able to produce substances with direct antiviral activity in their granules, such as eosinophil cationic protein (with ribonuclease activity), and they stimulate the antiviral immune response by acting as an antigen-presenting cell (APC) and enhancing the CD8+ T cell response (which is involved in antiviral immunity). In addition, eosinophils are able to express some Toll-like receptors (TLR) that are important for virus recognition, such as TLR-7, which binds to single-stranded RNA chains and recognizes single-stranded RNA (ssRNA) [24]. This receptor–ligand binding leads to an increase in adhesion molecules in eosinophils that decrease in peripheral blood and increase in tissues. In the lung, eosinophils can cause a significant inflammatory response through the production of superoxide anions and may be partially responsible for triggering an excessive immune response [24]. In COVID-19 patients, this mechanism may explain the correlation between eosinopenia and the cytokine storm that occurred in patients with extensive lung involvement and severe disease [25].

Regarding therapies, we found significant differences in treatment with corticosteroids and tocilizumab: most patients infected with the Delta variant were treated with corticosteroids and tocilizumab, whereas the use of these treatments in patients infected with the Omicron variant was significantly lower. The lower use of corticosteroids and tocilizumab in patients infected with the Omicron variant may be explained by the fact that the disease was milder during this period and fewer patients required oxygen therapy.

When comparing the two variants, we found, as expected, that more vaccinated patients visited ED in the second wave than in the first wave. These interesting data can be explained by the fact that approximately 90% of the population in our region was vaccinated during this period as part of the vaccination campaign, which increases the likelihood that COVID-19 infected patients will be vaccinated. The increase in vaccinated patients should not be interpreted as a vaccination gap, because if this were the case with a 90% vaccinated population, the prevalence of COVID-19-infected patients with severe disease would be 90%, which of course is not the case.

By comparing the two waves, we have seen how much things have changed in a short period of time, both from the perspective of the infected population and the prognosis and from the perspective of the virus. Therefore, validation of the scores developed in the first wave is needed. Several scores have been proposed and evaluated to assess mortality in patients with COVID-19. However, few of them have been studied considering the Omicron virus variants and vaccination status. A recent Thai study [26] evaluated the ability of some scores (REMS, NEWS, qSOFA, and MEWS) to assess mortality in a period when the Delta variant was prevalent. In this study of 987 patients, REMS showed the highest accuracy with an AUROC of 0.771 (0.738–0.804). However, the mortality rate observed in this study was considerably higher compared with our data (26% versus 13%). This discrepancy could be due to the recruitment system (we included all patients presenting to our ED, not only those admitted to the hospital) and the different levels of care in different hospitals. In addition, the authors did not consider the patients’ vaccination status, therapies performed, and concomitant diseases, which complicates the comparison between the two studies. In another study with a very large sample (more than 90,000 patients), the qSOFA, shock index, NEWS2, and quick COVID-19 severity index (qCSI) were evaluated at admission to ED to predict mortality [27]. In this case, the recorded mortality rate (17%) and inclusion criteria were similar to ours. However, none of the scores studied showed acceptable accuracy, with NEWS2 performing best (AUROC 0.593, 95%CI 0.588–0.597). Consequently, the authors agreed that the commonly used early warning scores for sepsis are inadequate for assessing the prognosis of COVID-19 patients and that the development of specific scores for COVID-19 is urgently needed. The 4C mortality score is the most commonly used and tested specific score to assess COVID-19-related mortality. During the first waves of COVID-19, it showed excellent accuracy in predicting mortality and has been validated in subsequent studies [10,28]. Recently its predictive utility was reported to decrease depending on the wave in which it was used. It showed an AUROC of 0.87 (0.78–0.94) in the first wave and 0.59 (0.33–0.84) in the second wave [29]. A more recent study demonstrated that the 4C mortality score also had good diagnostic accuracy in the Omicron wave, with an AUROC of 0.78 (0.74–0.82) [30]. However, vaccination status was not considered in this study.

To determine whether the ANCOC score we developed was good regardless of vaccination status and SARS-CoV-2 variants, we calculated the ROC curves and AUC for the entire study population (Figure 1), the Delta wave (Figure 2), and the Omicron wave (Figure 3). We found that the AUC was mostly above 0.8 for both the Delta and Omicron variants and for vaccinated and not vaccinated subjects, suggesting that the ANCOC score has very good accuracy in predicting deaths, regardless of vaccination status and variant considered. Further studies that also evaluate the number and type of vaccinations administered and the new therapies now available for COVID-19 are still needed.

Our study has some limitations. First, it is a retrospective study with limitations related to its design (selection bias). In addition, it is a monocentric study, and generalization of the data is not possible because of the lack of external validation. On the other hand, an error related to sampling variability between different laboratories was avoided. In addition, the viral variant data were not examined directly, but using data from the national health agency (however, the prevalence in Italy of the Delta and Omicron variants was approximately 99% during the collection periods) [14,15].

## 5. Conclusions

We found that the ANCOC score could be a useful guide to accurately determine a patient’s risk of death independent of viral variants, therapies, and vaccination status. The ANCOC score is simple, uses a small number of parameters, all of which are routinely assessed at ED in patients with COVID-19, and can help emergency physicians decide whether or not to admit the patient. However, further studies are needed to externally validate our results before they can be generalized to other populations and settings.

## Figures and Tables

**Figure 1 jcm-12-05838-f001:**
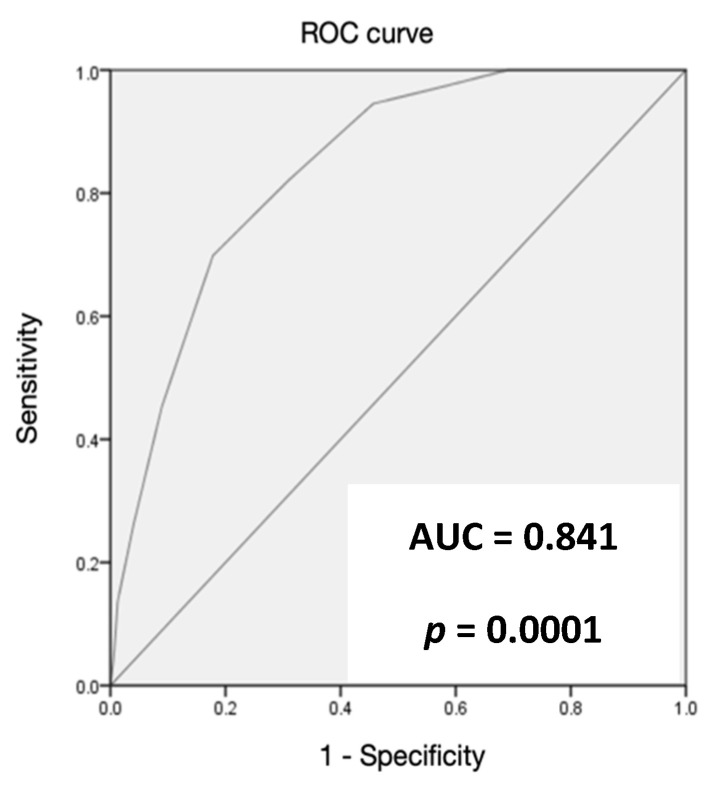
The area under the receiver operating characteristic (AUROC) curve analysis for ANCOC as a predictor of mortality (whole population).

**Figure 2 jcm-12-05838-f002:**
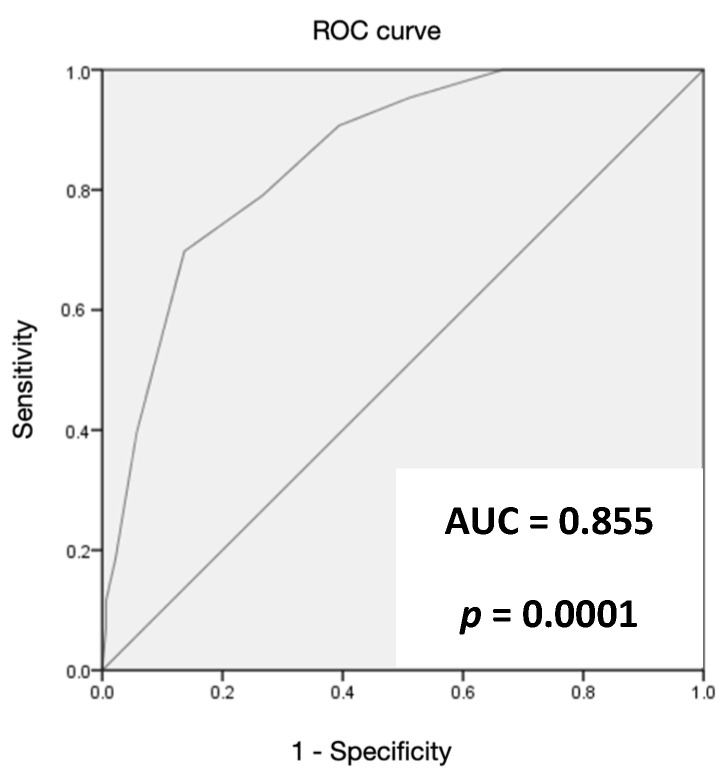
The area under the receiver operating characteristic (AUROC) curve analysis for ANCOC as a predictor of mortality (Delta wave).

**Figure 3 jcm-12-05838-f003:**
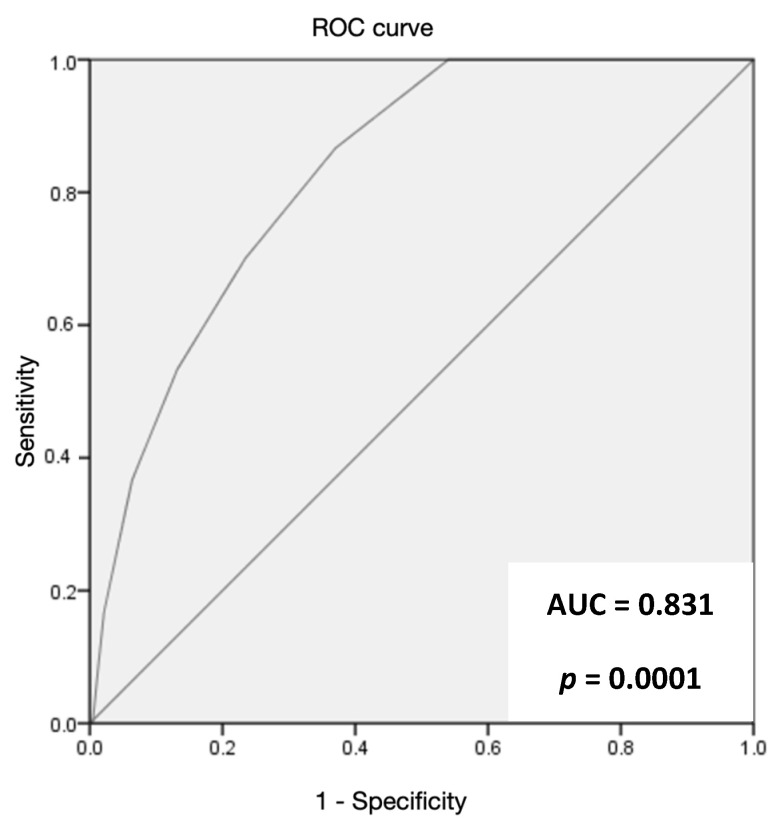
The area under the receiver operating characteristic (AUROC) curve analysis for ANCOC as a predictor of mortality (Omicron wave).

**Figure 4 jcm-12-05838-f004:**
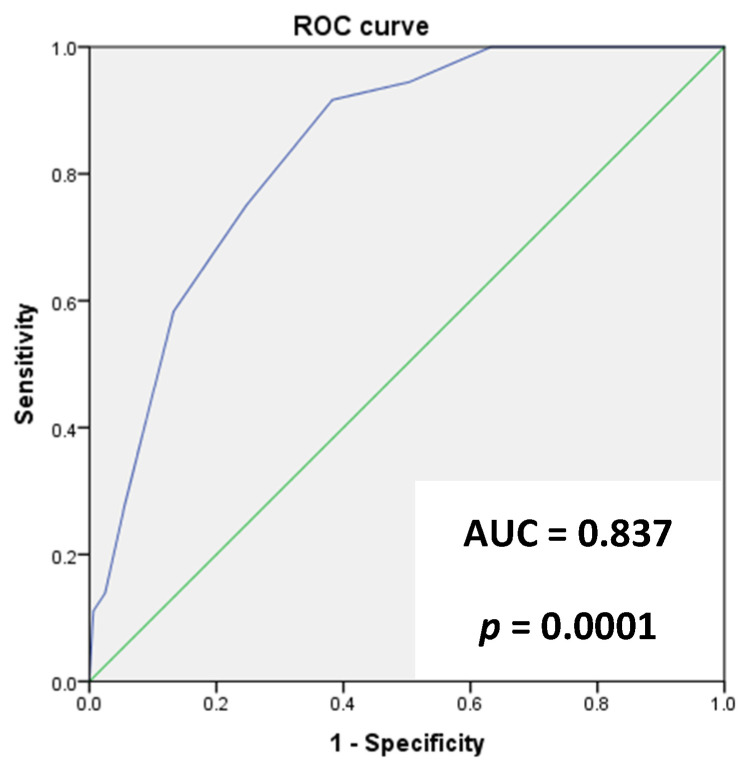
The area under the receiver operating characteristic (AUROC) curve analysis for ANCOC as a predictor of mortality in non-vaccinated patients.

**Figure 5 jcm-12-05838-f005:**
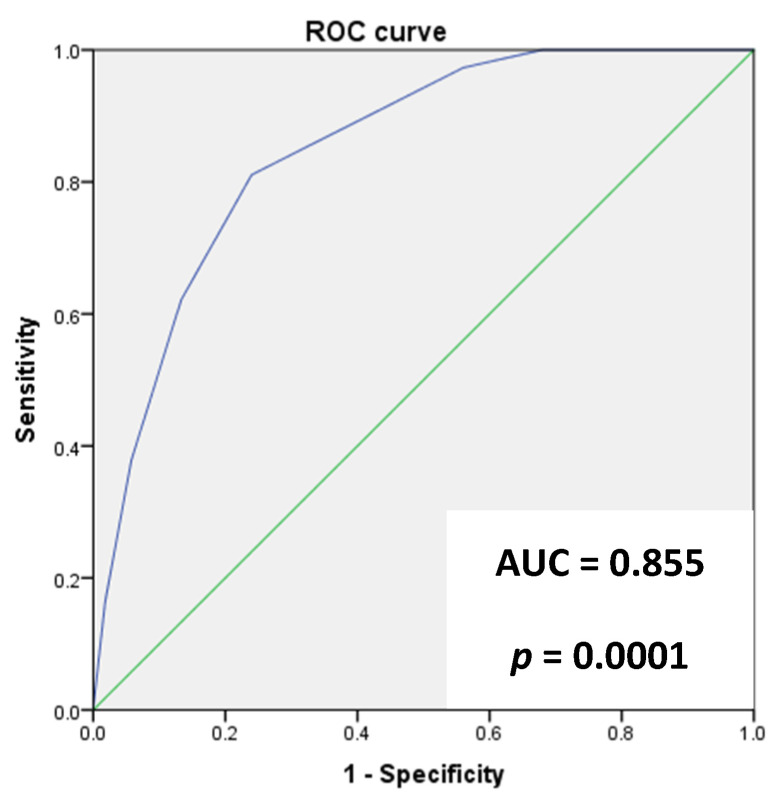
The area under the receiver operating characteristic (AUROC) curve analysis for ANCOC as a predictor of mortality in vaccinated patients (at least one dose).

**Table 1 jcm-12-05838-t001:** ANCOC score.

Criterion	Score
BUN > 35	1
BUN < 15	−1
SO_2_ > 96	−1
SO_2_ ≤ 88	1
Comorbidities < 2	−1
Comorbidities > 4	1
Age < 55	−2
Age > 80	2
CRP > 26	−1
CRP > 155	1

BUN: blood urea nitrogen, SO_2_: oxygen saturation, CRP: C-reactive protein.

**Table 2 jcm-12-05838-t002:** General information of all enrolled patients. Comparison between the patients infected by the Delta variant and those infected by the Omicron variant.

Clinical Data—Mean ± SD or n (%)	Total Population (843)	Delta (515)	Omicron 1 (328)	*p*-Value
Age (years)	62 ± 19	60 ± 18	64 ± 18	0.001
Male	481 (57)	287 (56)	194 (59)	0.3
Female	362 (43)	228 (44)	134 (41)	0.3
BMI (kg/m^2^)	27.3 ± 6	27.4 ± 7	27.1 ± 6	0.6
No comorbidities	265 (31)	170 (33)	95 (29)	0.31
One comorbidity	218 (26)	145 (28)	73 (22)	0.2
≥2 comorbidities	360 (43)	200 (39)	160 (48)	0.2
COI	64 ± 56	58 ± 54	72 ± 57	0.005

SD: standard deviation; n: number; BMI: body mass index, COI: cutoff index.

**Table 3 jcm-12-05838-t003:** Prevalence of comorbidities in enrolled patients (number and percentage).

Comorbidity n (%)	All Patients (843)	Delta Variant (515)	Omicron Variant (328)	*p*
Hypertension	320 (38)	175 (34)	145 (44)	0.003
Obesity	205 (24)	141 (27)	64 (20)	0.01
Diabetes	122 (14)	73 (14)	49 (15)	0.75
Active neoplasm	105 (12)	57 (11)	48 (15)	0.12
Atrial fibrillation	82 (10)	52 (10)	30 (9)	0.65
COPD	80 (9)	42 (8)	38 (12)	0.01
CHD	79 (9)	43 (8)	36 (11)	0.20
CKD	61 (7)	25 (5)	36 (11)	0.001
Heart failure	59 (7)	36 (7)	23 (7)	0.99
Other chronic disease	164 (19)	92 (18)	71 (22)	0.17

COPD: chronic obstructive pulmonary disease; CHD: coronary heart disease; CKD: chronic kidney disease. Other chronic diseases: Alzheimer’s disease, asthma, Parkinson’s disease, valvular heart disease, autoimmune disease, inflammatory bowel disease, chronic liver disease.

**Table 4 jcm-12-05838-t004:** Odds ratios and 95% confidence intervals for comorbidities significantly associated with Delta and Omicron variants’ differences after multiple logistic regression and correction for sex, age, and comorbidities.

	*p*-Value	Odds Ratio	95% CI
Obesity	0.26	0.72	0.41–1.27
Hypertension	0.01	0.48	0.28–0.84
COPD	0.01	0.40	0.19–0.83
CKD	0.70	0.83	0.31–2.17

CI: confidence interval; COPD: chronic obstructive pulmonary disease; CKD: chronic kidney disease.

**Table 5 jcm-12-05838-t005:** Oxygen saturation and PaO_2_/FiO_2_ of all enrolled patients. Comparison between the patients infected by the Delta variant and those infected by the Omicron variant.

	Total Population (843)	Delta (515)	Omicron 1 (328)	*p*-Value
O_2_ saturation (%)	94 ± 6	93 ± 6	95 ± 5	0.0001
PaO_2_/FiO_2_ (mmHg)	306 ± 111	296 ± 109	324 ± 115	0.001

SD: standard deviation, PaO_2_: partial pressure of oxygen in arterial blood, FiO_2_: inspired oxygen fraction.

**Table 6 jcm-12-05838-t006:** Laboratory parameters of all enrolled patients. Comparison between the patients infected by the Delta variant and those infected by the Omicron variant. All data are expressed as mean ± SD.

Laboratory Parameters Mean ± SD	Total Population	Delta	Omicron 1	*p*-Value
BUN (mg/dl)	26 ± 24	24 ± 23	29 ± 25	0.005
LDH (U/L)	320 ± 198	340 ± 184	288 ± 214	0.005
Eosinophils (cells × 10^9^/L)	0.05 ± 0,12	0.04 ± 0.12	0.07 ± 0.12	0.005
Lymphocytes (cells × 10^9^/L)	3.1 ± 3.7	4.0 ± 4.7	1.6 ± 3.8	0.25
Fibrinogen (mg/dl)	478 ± 169	513 ± 171	421 ± 149	<0.001
Procalcitonin (ng/mL)	1.8 ± 20	1.60 ± 22	2.10 ± 14	0.72
D-dimer (ng/mL)	2589 ± 5623	2488 ± 5788	2787 ± 5293	0.51
CRP (mg/dL)	67 ± 71	69 ± 69	64 ± 74	0.36

SD: standard deviation; BUN: blood urea nitrogen; LDH: lactate dehydrogenase; CRP: C-reactive protein.

**Table 7 jcm-12-05838-t007:** Pharmacological therapy and oxygen supplementation therapy of all enrolled patients. Comparison between the patients infected by the Delta variant and those infected by the Omicron variant.

Treatment n (%)	Total Population (843)	Delta (515)	Omicron 1 (328)	*p*-Value
Remdesivir	194 (23)	139 (27)	55 (16)	0.005
Corticosteroids	508 (60)	361 (70)	147 (45)	0.0001
Anti-IL6R	105 (12)	113 (22)	13 (4)	0.001
Mab	85 (10)	48 (9)	37 (11)	0.35
Anticoagulants	697 (83)	434 (84)	263 (80)	0.64
HFNT	117 (14)	88 (17)	29 (9)	0.001
NIV	80 (9)	60 (11)	20 (6)	0.01
OTI/Tracheostomy	65 (7)	6 (7)	29 (9)	0.3

Mab: monoclonal antibodies; IL6R: interleukin 6 receptor; HFNT: high-flow nasal therapy; NIV: non-invasive ventilation; OTI: orotracheal intubation.

**Table 8 jcm-12-05838-t008:** Outcome of all enrolled patients. Comparison between the patients infected by the Delta variant and those infected by the Omicron variant.

Outcome n (%)	Total Population (843)	Delta Population (515)	Omicron 1 Population (328)	*p*-Value
Discharged from ED	191 (23)	79 (15)	112 (34)	0.0001
Admitted in medical ward	509 (60)	336 (65)	173 (53)	0.0003
Admitted in ICU	143 (17)	100 (20)	43 (13)	0.02
Deceased	114 (13)	73 (14)	41 (12)	0.48

ED: emergency department, ICU: intensive care unit.

**Table 9 jcm-12-05838-t009:** Vaccination status of all enrolled patients. Comparison between the patients infected by the Delta variant and those infected by the Omicron variant.

Vaccination Status	Total Population	Delta Population	Omicron 1 Population	*p*-Value
Vaccinated n (%)	357 (42)	176 (34)	183 (56)	0.001
Not vaccinated n (%)	486 (58)	339 (66)	145 (44)	0.001

## Data Availability

Data are available on reasonable request to the corresponding author.

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
