# Peer review of "ANCOC Score to Predict Mortality in Different SARS-CoV-2 Variants and Vaccination Status"

_jcm, 2023, doi:10.3390/jcm12185838_

Round 1
Reviewer 1 Report
ABSTRACT
- The conclusion mentioned ICU admission, which the text did not elaborate on. I suggest the author keep track of mortality instead of other outcomes.
INTRODUCTION
- The author mentioned several current scoring systems. Are there any reports regarding the usage of these scores in the real world, and not limited to research settings only?
METHODS
- Was there any calculation for the sample size?
- Was the sample set different from the original set that was used for the development of the score?
- For the multivariate analysis, was there any particular reason why the author used a p-value of 0.1 as the limit instead of 0.2 or 0.25?
RESULTS
- I suggest the author add the age range and the BMI range of the subjects
- Was there any data regarding other antiviral treatments than remdesivir in the medical record?
- Did the subjects use the same vaccines?
- Was there any data regarding the difference between one, two, or three doses of vaccines?
DISCUSSION
- Was there any explanation regarding the eosinophils as the significant risk factor in the univariate analysis?
REFERENCES
- I suggest the author add the English translation for the title of non-English language references (two of them).
FIGURES
- The title of Figure 2 was the same as that of Figure 3.
Author Response
Reviewer 1
Answer: Thank you for taking the time to read and comment on our paper, and for the valuable suggestions that have enabled us to improve it considerably
ABSTRACT
- The conclusion mentioned ICU admission, which the text did not elaborate on. I suggest the author keep track of mortality instead of other outcomes.
Answer: We deleted the sentence related to the ICU admission
INTRODUCTION
2) The author mentioned several current scoring systems. Are there any reports regarding the use of these scores in the real world, and not limited to research settings only?
Answer: As far as we know, there is no real-world data about the use of these scores. We added a sentence in the introduction on this issue.
METHODS
- Was there any calculation for the sample size?
Answer: The sample size was calculated to determine the estimated Auroc value of at least 0.8 for the ANCOC score. Considering the 15.2% mortality observed in our previous study*, a width of the confidence interval of 0.150, a confidence interval of 0.95, an alpha error of 0.05, and a power of 80% we calculated a minimum sample size of 348 patients.
We add this information to the text
* Candelli M. et al, A new simple score to predict mortality of COVID-19 in the emergency department. Signa Vitae. 2023; 19(2): 20-27. doi: 10.22514/sv.2022.025.
4) Was the sample set different from the original set that was used for the development of the score?
Answer: The sample size was set differently because this time we were able to use data on mortality from our previous study
5)For the multivariate analysis, was there any particular reason why the author used a p-value of 0.1 as the limit instead of 0.2 or 0.25?
Answer: As there is no standard cut-off for determining the choice of variables, we used a cut-off value of p<0.1 because it seemed to us a good compromise between the risk of including confounding variables and that of excluding variables actually correlated to the outcome. In addition, this cut-off allowed us to select a number of variables suitable for the sample size
RESULTS
6) I suggest the author add the age range and the BMI range of the subjects
Answer: We add a range for age and BMI in the text
7) Was there any data regarding other antiviral treatments than remdesivir in the medical record?
Answer: Because paxlovide was not approved in Italy until late January 2022, no patients were treated with it. Few participants in either group were treated with molnupiravir. We have included these data in the text
8) Did the subjects use the same vaccines?
Answer: Most patients (83%) received the mRNA-BNT162b2 vaccine, 11% received the mRNA-1273 vaccine, and 3% received both. For the remaining patients (3%), we did not have information on the type of vaccine used. We have included these data in the text
9) Was there any data regarding the difference between one, two, or three doses of vaccines?
Answer: Data on patient subgroups were not analyzed. The number of patients with a single injection and the number of patients with three injections was too small, so it seemed better not to provide data only for the group with two injections.
DISCUSSION
10) Was there any explanation regarding the eosinophils as the significant risk factor in the univariate analysis?
Answer: We add an explanation of the phenomenon in the discussion
REFERENCES
11) I suggest the author add the English translation for the title of non-English language references (two of them).
Answer: Thanks for the suggestion. We added the translation.
FIGURES
12) The title of Figure 2 was the same as that of Figure 3.
Answer: We add the correct title of Figure 3
Reviewer 2 Report
Dear authors
In the current retrospective research, the authors aimed to validate the ANCOC (age, blood urea nitrogen, C-reactive protein, oxygen saturation, comorbidities) clinical score as a prognostic score and if it can predict the 60-day mortality risk independent of vaccination status and viral variant. In spite of the declaration of the pandemic, this point is very important and interesting.
I have a few comments on this manuscript:
Abstract:
· Numbers of patients infected with the Delta and Omicron-1 variants are unclear.
Introduction:
· Many abbreviations were mentioned without complete names, such as NEWS, NEWS2, NEWS-C, MEWS, qSOFA, REMS, ED, IRCCS …etc) and the reference used did not discuss all these scores.
· The main protocol " International Severe Acute Respiratory and Emerging Infections Consortium (ISARIC) World Health Organization (WHO) Clinical Characterisation Protocol UK (CCP-UK) study" is not mentioned.
Method:
· The numbers of patients in the two groups (depending on which variant Delta or Omicron 1) are not clear.
· What is the basis for choosing -1, 1, 2, or -2 in the score??
Results:
· The author mentioned that "the period in which the Delta variant was over-Riding" is this mean these patients infected with this variant or there is a clear sharp diagnosis? The same with the other variant.
· In the vaccination status, which vaccine was used?
Discussion:
· It is very short and lacks comparative studies and clarification of the results.
References:
· Authors in reference 11 are not mentioned.
· References 11, and 12 are not complete.
Regards,
Author Response
Reviewer 2
Dear authors
- In the current retrospective research, the authors aimed to validate the ANCOC (age, blood urea nitrogen, C-reactive protein, oxygen saturation, comorbidities) clinical score as a prognostic score and if it can predict the 60-day mortality risk independent of vaccination status and viral variant. In spite of the declaration of the pandemic, this point is very important and interesting.
Answer: We thank you for your appreciation of the study, for taking the time to review it, and for the advice that helped us improve the quality of the study.
I have a few comments on this manuscript:
Abstract:
2) The numbers of patients infected with the Delta and Omicron-1 variants are unclear.
Answer: We added how many patients had the delta variant and how many had the Omicron-1 variant
Introduction:
3) Many abbreviations were mentioned without complete names, such as NEWS, NEWS2, NEWS-C, MEWS, qSOFA, REMS, ED, IRCCS … etc.) and the reference used did not discuss all these scores.
Answer: We mentioned the complete names of all scores and abbreviations. NEWS, NEWS2, NEWS-C, MEWS, qSOFA, and REMS were discussed in reference 5, the C4 score in reference 6 and 7, and GOL2D score in reference 8
4) The main protocol " International Severe Acute Respiratory and Emerging Infections Consortium (ISARIC) World Health Organization (WHO) Clinical Characterisation Protocol UK (CCP-UK) study" is not mentioned.
Answer: We add the reference for the main protocol
Method:
5) The numbers of patients in the two groups (depending on which variant Delta or Omicron 1) are not clear.
Answer: We added how many patients had the delta variant and how many had the Omicron-1 variant
6) What is the basis for choosing -1, 1, 2, or -2 in the score??
Answer: In this paper, we simply applied the score developed in the previous study to test accuracy in the presence of different viral variants and vaccination status. Scores were selected in the previous study based on cut-off values with maximum negative predictive value (-1) and maximum positive predictive value (+1) of each variable for mortality. A double score (+2 or -2) was used for age because it showed a stronger (more than double) correlation with mortality (measured as OR) than the other parameters in the multivariate analysis.
Results:
7) The author mentioned that "the period in which the Delta variant was over-Riding" is this mean these patients infected with this variant or there is a clear sharp diagnosis? The same with the other variant.
Answer: As stated in the discussion data on viral variant have not been investigated directly but with data from the national health agency (however, during the enrollment periods prevalence of delta and omicron variant in Italy were around 99%).
8) In the vaccination status, which vaccine was used?
Answer: We added data on vaccines used
Discussion:
9) It is very short and lacks comparative studies and clarification of the results.
As suggested, we improved the discussion.
References:
10) Authors in reference 11 are not mentioned.
11) References 11, and 12 are not complete.
Answer: References 11 and 12 are from the AIFA (Drugs Italian Agency) and ISS (Italian National Institute of Health) websites, so it was not possible to indicate the author. We follow the rules for citing websites as in other articles already published in JCM.
For example the reference number 9 (Tracking SARS-CoV-2 Variants. Available online: https://www.who.int/activities/tracking-SARS-CoV-2-variants (accessed on 22 January 2023) of the paper Dobrowolska, K. et al. Retrospective Analysis of the Effectiveness of Remdesivir in COVID-19 Treatment during Periods Dominated by Delta and Omicron SARS-CoV-2 Variants in Clinical Settings. J. Clin. Med. 2023, 12, 2371. https://doi.org/10.3390/jcm12062371. We have added an Italian translation of the documents.
Reviewer 3 Report
Dear Authors, I have read your manuscript with interest.
The current manuscript titled: "ANCOC score to predict mortality in different SARS-CoV-2 variants and vaccination status" represents an important analysis of evolving field of Infectious Diseases, Internal Medicine and Immunology.
In my opinion, these are the adjustments which should be made to increase the value of your manuscript:
1. In Abstract, please, add abbreviations for “SARS-CoV-2”, “COVID-19”, “ANCOC”, “COPD”.
2. In Introduction chapter, please, add more detailed information about COVID-19, inclusive epidemiology. Also, add information about all pandemic waves and vaccination. It is recommended to study the following recently published article: https://doi.org/10.3390/diagnostics12061373.
3. Line 79: change “COVID” to “COVID-19”.
4. Line 82: change “SARS-CoV2” to “SARS-CoV-2.”
5. In the section describing the patients’ treatment, there is no information about antibiotic ant anticoagulant therapy. Please describe this medication in detail.
6. In the Discussion section, there is not enough comparative information with other studies.
7. In Conclusions section please highlight the practical implications of your study and its relevance to clinical practice.
8. Add future perspectives.
9. The manuscript contains some punctuation errors (e.g., lines 20, 23, 52, 94, 95, 271, 273, 278, etc.), please revise the text.
Minor editing of English language required
Author Response
Reviewer 3
Dear Authors, I have read your manuscript with interest.
- The current manuscript titled: "ANCOC score to predict mortality in different SARS-CoV-2 variants and vaccination status" represents an important analysis of the evolving field of Infectious Diseases, Internal Medicine, and Immunology.
Answer: We thank the reviewer for liking our study and for helping to improve it significantly with his suggestions. Thanks for taking the time to evaluate the study
In my opinion, these are the adjustments which should be made to increase the value of your manuscript:
- In the Abstract, please, add abbreviations for “SARS-CoV-2”, “COVID-19”, “ANCOC”, and “COPD”.
Answer: We have added abbreviations for “SARS-CoV-2”, “COVID-19”, “ANCOC”, and “COPD”.
- In the Introduction chapter, please, add more detailed information about COVID-19, inclusive epidemiology. Also, add information about all pandemic waves and vaccinations. It is recommended to study the following recently published article: https://doi.org/10.3390/diagnostics12061373.
Answer: We have improved the introduction as suggested by the reviewer
- Line 79: change “COVID” to “COVID-19”.
Answer: We have corrected the error
- Line 82: Change “SARS-CoV2” to “SARS-CoV-2.”
Answer: We have corrected the error
- In the section describing the patients’ treatment, there is no information about antibiotic and anticoagulant therapy. Please describe this medication in detail.
Answer: We did not collect data on antibiotic treatment. Data on anticoagulants were added to the paper
.6. In the Discussion section, there is not enough comparative information with other studies.
Answer: We improved the section and added comparative information with other studies
- In the Conclusions section please highlight the practical implications of your study and its relevance to clinical practice.
Answer: We have changed the conclusions according to the reviewer’s suggestion
- Add future perspectives.
Answer: We have changed the conclusions and discussion according to the reviewer’s suggestion
- The manuscript contains some punctuation errors (e.g., lines 20, 23, 52, 94, 95, 271, 273, 278, etc.), please revise the text.
Answer We have fixed punctuation errors and improved the quality of the English language
Comments on the Quality of English Language
Minor editing of the English language required
Answer: We have fixed punctuation errors and improved the quality of the English language
Round 2
Reviewer 3 Report
I agree with the changes made, which significantly improve the quality of the manuscript.
Minor editing of English language required
Author Response
Thank you for your comments, which have allowed us to improve the quality of our work. We have thoroughly corrected the English language, particularly some punctuation and spelling errors.